# Reasons for Altering Bladder Management and Satisfaction with Current Bladder Management in Chronic Spinal Cord Injury Patients

**DOI:** 10.3390/ijerph192417032

**Published:** 2022-12-18

**Authors:** Hui-Ling Yeh, Hann-Chorng Kuo, Chuan-Hsiu Tsai, Ru-Ping Lee

**Affiliations:** 1Department of Nursing, Hualien Tzu Chi Hospital, Buddhist Tzu Chi Medical Foundation, Hualien 970473, Taiwan; 2Institute of Medical Sciences, Tzu Chi University, Hualien 970374, Taiwan; 3Department of Urology, Hualien Tzu Chi Hospital, Buddhist Tzu Chi Medical Foundation, Hualien 970473, Taiwan; 4Department of Nursing, Tzu Chi University, Hualien 970374, Taiwan

**Keywords:** spinal cord injury, bladder management, satisfaction, urinary tract infection

## Abstract

Patients with spinal cord injury (SCI) often require bladder management. However, patients routinely change their bladder management for better satisfaction. The reasons for altering a bladder management method in SCI patients remain insufficiently understood. The purposes of this study are to assess current satisfaction with bladder management and the reasons for changing bladder management in SCI patients. A prospective cross-sectional survey with a convenience sampling method was used. The study was conducted from January 2018 to December 2019. The inclusion criteria included an age ≥18 years and a diagnosis of SCI more than one year previously. The questionnaires were self-administered and collected from eligible patients during a free clinic service. A total of 515 SCI participants were enrolled. Two hundred and eighty-three (55.0%) participants had experienced changing their bladder management. The most used method of current bladder management was self-voiding. About 84.7% of participants reported being satisfied with their current bladder management. Bladder management changes were most often made due to frequent urinary tract infections. Furthermore, the participants dissatisfied with their management had more urological complications. This study indicates that appropriate bladder management can improve the subjective satisfaction of patients. For long-term care, preventing urinary tract infections is a helpful strategy for patients’ satisfaction with bladder management.

## 1. Introduction

Chronic spinal cord injury (SCI) often leads to neurogenic lower urinary tract dysfunction (NLUTD), which strongly influences normal bladder storage and urination [1]. NLUTD involves urinary incontinence caused by detrusor overactivity (DO), incomplete bladder emptying, impaired bladder compliance, detrusor sphincter dyssynergia, autonomic dysreflexia, and detrusor areflexia [2], and it can lead to recurrent urinary tract infections (UTI), repeated stone formation, hydronephrosis, vesicoureteral reflux, urosepsis, and renal function insufficiency. If end-stage renal failure or uremia develops, the patient must undergo lifetime hemodialysis [3,4]. NLUTD can be treated through appropriate bladder management, which prevents urological complications, facilitates bladder emptying, preserves kidney function, and improves the subjective satisfaction of patients with SCI [5]. By contrast, inappropriate bladder management can increase the likelihood of hospitalization and result in severe complications [1,3,4,6].

Appropriate bladder management can reportedly increase patient satisfaction [2]. Clean intermittent catheterization (CIC) is the preferred method of treating NLUTD in patients with chronic SCI [5]. However, the ideal bladder management is not necessarily suitable for all patients with SCI because lower urinary tract dysfunction varies among patients with different levels of SCI. The development of urological complications may also cause frequent changes in a patient’s lifestyle, economic situation, personal preference, and primary caregiver’s burden [7]. In addition, lower urinary tract dysfunction can change over time, resulting in differences in storage and voiding dysfunction over the long disease period. Bladder management should be modified in accordance with changes in lower urinary tract function; otherwise, urological complications, such as recurrent UTI, upper tract deterioration, and urinary incontinence, may worsen as the disease progresses. However, the reasons for altering a bladder management method and patient satisfaction with bladder management for SCI remain insufficiently understood. Therefore, the aim of this study is to assess patient satisfaction with current bladder management and the reasons for altering the bladder management method in patients with chronic SCI.

## 2. Materials and Methods

### 2.1. Subsection

This study involved a prospective cross-sectional survey conducted between January 2018 and December 2019 through a nationwide free clinic service. A convenience sampling method was used. This study used the Strengthening the Reporting of Observational Studies in Epidemiology (STROBE) Statement as a guide.

### 2.2. Participants and Settings

Free clinic surveillance along with a health education program was conducted in 12 cities in Taiwan, with a total of 14 free clinics included. Prior to the free clinic, we contacted the regional SCI association of each city. The inclusion criteria for participants were age ≥ 18 years and a diagnosis of SCI 1 year or more prior. Written informed consent was obtained from each participant, and a self-administered questionnaire was distributed to each eligible patient before or during the free clinic service (Figure 1). The study was approved by the Institutional Review Board and Ethics Committee of Hualien Tzu Chi Hospital, Buddhist Tzu Chi Medical Foundation (IRB106-125-B). The authors confirm all patient identifiers have been removed or disguised so the patients described are not identifiable and cannot be identified through the details of the study.

### 2.3. Outcome Measurements

One researcher assisted the patients in answering the questionnaire. The questionnaire included questions on patient demographics, initial and current bladder management, types of urological procedures or surgeries received, previous urological complications, and satisfaction with current bladder management. The methods for bladder management were the following: (1) self-voiding through abdominal straining, stimulation, or reflexion (incontinence); (2) an indwelling catheter (IDC) through the urethra or suprapubic cystostomy; (3) CIC by the patient themselves or a caregiver; and (4) other. The urological procedures and surgeries included suprapubic cystostomy, transurethral incision of the bladder neck, transurethral incision or resection of the prostate, botulinum toxin injection, external sphincterotomy, ureteral reimplantation, ileal conduit, continent urinary diversion, and augmentation enterocystoplasty. The urological complications were UTI, acute pyelonephritis, dysuria, urinary incontinence, urinary tract stones, urethral stricture, urinary tract fistula, acute epididymitis, hydronephrosis, autonomic dysreflexia, and vesicoureteral reflux.

A self-administered questionnaire was developed and distributed to each eligible patient. Five medical experts validated the questionnaire, which had a total average content validity index of 0.97. Satisfaction was evaluated by one global rating scale on satisfaction, and the scale was divided into “Not at all Satisfied”, “Partly Satisfied”, “Satisfied”, “More than Satisfied”, and “Very Satisfied”, categories. The urological complications that were evaluated included 11 questions for our patients to indicate whether they have ever occurred. Change of bladder management was defined as changing from their initial bladder management more than once after being given the diagnosis of SCI. 

### 2.4. Statistical Analysis

The patient characteristics are presented as percentages or means and standard deviations. Continuous variables were evaluated using Student’s *t*-test, and categorical variables were assessed using the chi-square test. The significance of the differences in satisfaction with bladder management changes was analyzed using the independent sample *t*-test. The Kolmogorov–Smirnov test was performed to evaluate the normal distribution of all the measurement variables. All statistical data were analyzed using SPSS for Windows, version 22.0 (Armonk, NY, USA). Significance was assessed at the 0.05 level, and all tests were two-tailed.

## 3. Results

### 3.1. Participant Characteristics

In total, 515 patients with SCI were included, with a mean age of 52.6 ± 13.9 years and a male proportion of 77.3%. The mean injury age was 32.8 ± 14.5 years, and the mean duration since injury was 19.3 ± 12.7 years. In addition, 44.1% of the patients were married, and 40.4% were unmarried. Cervical spine injury accounted for 45.6% of the patients’ injuries. In total, 48.2% of the patients were grade A on the American Spinal Injury Association (ASIA) impairment scale, and most exhibited paraplegia (59.2%). At the time of this study, 47.6% of the participants could empty their bladders through any means of self-voiding, 32.4% required an IDC, and 15.5% required CIC to empty their bladders. Among the 515 patients, 329 (63.9%) had undergone urology surgery, and 425 (82.5%) had experienced urology complications (Table 1).

### 3.2. Participants’ Satisfaction with Their Bladder Management

The participants were divided into two groups in accordance with their satisfaction with their current bladder management (Table 2). Among the 515 patients, 436 (84.7%) were satisfied with their current bladder management and 79 (15.3%) were dissatisfied. No significant differences were identified between the two groups with respect to sex, injury age, injury years, marital status, injury level, ASIA impairment grade, limb status, current bladder management, or previous urological surgery. However, the dissatisfied group had a significantly higher average age (*p* = 0.026) and more urological complications (*p* = 0.005) than the satisfied group. Data regarding the participants’ satisfaction and dissatisfaction with their bladder management are listed in Table 2.

### 3.3. The Reason for Changing Bladder Management

We further divided the participants into two groups on the basis of whether they had had a change in bladder management (Table 3). Overall, 283 (55.0%) had had a change in bladder management. Sex and injury level were similar between the groups. In addition, the participants who had had their bladder management changed were generally younger at the time of the interview (*p* < 0.001) and the time of injury (*p* = 0.003) and had a shorter SCI disease duration (*p* = 0.004). Patients who had had a change in bladder management were more likely to be unmarried (*p* = 0.040) and have tetraplegia (*p* = 0.001). In the change in bladder management group, there were more likely to have ASIA-A and ASIA-B impairment (73.5%). The difference in the ASIA impairment grade between the groups was significant (*p* = 0.003). In addition, the differences in initial bladder management (*p* < 0.001) and current bladder management (*p* < 0.001) were significant. The participants who could urinate spontaneously by any means generally did not require a change in bladder management, whereas those who employed IDC or CIC had had a change in bladder management (*p* < 0.001). Furthermore, 329 patients (63.9%) had a history of urological surgery, and 425 patients (82.5%) exhibited urological comorbidities. The group with a change in bladder management were significantly more likely to have had urological surgery (*p* < 0.001) and urological complications (*p* < 0.001) than the group without a change. Furthermore, 329 patients (63.9%) had a history of urological surgery, and 425 patients (82.5%) exhibited urological comorbidities. Nevertheless, the overall satisfaction with bladder management was similar between the groups (88.4% vs. 81.6%, *p* = 0.195; Table 3).

Of the patients in the initial CIC group, seventeen of them switched to the IDC group due to recurrent UTI (15, 88%) and lack of help (2, 12%). One hundred and ninety people were in the IDC group in the first 6 months, whereas 114 (60%) remained in the IDC group, 29 (15.3%) changed to the CIC group, and 45 (23.7%) regained control of self-voiding. Sixty-eight (40.7%) of the current IDC group had undergone suprapubic cystostomy surgery (Figure 2). The reasons for the changes were recurrent infection (24, 34.5%), care related factors (26, 38%), and personal factors (18, 26%) (Figure 2).

The main reason for changing bladder management was frequent UTI (51%), followed by patients choosing the bladder management that they think is more suitable (31%), poor caregiver cooperation (14%), and complications, such as newly developed hydronephrosis (9%), voiding problems caused by dysuria or urine leakage (7%), and impaired renal function (7%) (Figure 3).

## 4. Discussion

This is the first study to investigate patients’ satisfaction with their bladder management and the causes of changing bladder management in patients with chronic SCI through free clinic surveillance in Asia. Among the 515 participants, more than 80% were satisfied with their current bladder management. The preferred method of bladder management was self-voiding. Furthermore, the participants who could urinate spontaneously after injury generally did not require a change in bladder management, and those who had had a bladder management change experienced more urological complications and surgeries. However, the bladder-emptying method did not influence self-reported satisfaction.

The most common method of bladder management for patients with SCI who cannot void completely is CIC [8,9,10]. Although CIC is beneficial to patients with SCI, most patients consider it to be inconvenient and, therefore, discontinue CIC and seek alternative methods of bladder emptying [11]. In the current study, the patients with SCI mostly preferred self-voiding for bladder management. Approximately 47% of the patients maintained self-voiding almost 20 years after injury. However, according to Zlatev’s report, 76% of patients with SCI cannot void volitionally and require CIC [10]. This difference between the two studies may be related to differences in patient enrollment, overall physical condition, and injury duration. We enrolled patients through free clinic surveillance, and, therefore, included participants with better physical condition and longer injury duration than may have been enrolled in other studies. Patients with SCI who had severe disability and used IDCs or required CIC may not have attended the clinic surveillance.

In this study, 55% of the participants had changed from their initial bladder management at least once after injury. This rate of changing bladder management is similar to that reported by Savic et al. (50.6%) in the United Kingdom [11]. However, the rate of satisfaction with current bladder management in this study was high, regardless of whether the participants had undergone a change in bladder management. This may be because the population in this study had a longer mean injury time (19.3 years). Furthermore, the participants in this study were generally young, were unmarried, had ASIA-A impairment, and were willing to change their initial bladder management through additional urological surgery and to seek more favorable means of urination. Additionally, for the patients who had had a change in bladder management, their initial and current bladder management methods were similar, suggesting that, although the participants attempted to change their management, they ultimately selected their original bladder management method as their final management. For example, patients with neurogenic DO may be eager to receive detrusor botulinum toxin A injections to make them dry. However, CIC can be difficult to maintain due to the inconvenience of public restrooms; this may cause patients to return to self-voiding. Due to uncompleted emptying, participants need to switch to IDC to avoid recurrent UTI and protect renal function. 

Moreover, in this study, 62.5% of patients with SCI with paraplegia and 27.6% with tetraplegia underwent a change in bladder management to treat urinary incontinence and upper urinary tract deterioration. The patients with paraplegia generally had good hand function and were seeking more appropriate bladder management or treatment of urological complications to improve their quality of life. Patients with tetraplegia are often highly dependent on their primary caregivers to empty their bladders after surgical bladder management. Patients with SCI often report a need to change their bladder management. One study reported that patients with initial CIC may change their bladder management to less optimal strategies, such as IDC [6]. Our study reveals a similar phenomenon, in which CIC participants decided to switch to IDC due to recurrent UTI. For all forms of bladder management, the strategy employed for patients with SCI should be individualized, and the patient’s wishes, family support, and the level of convenience in performing medical treatments should be considered [7].

SCI is a long-term chronic disease. Therefore, age should be considered when selecting bladder management. Singh reported that as the age of patients with SCI increases, their catheter retention rate increases [12]. However, in our cohort study, the IDC retention rate decreased from 36.9% to 32.4% over the 20 years of follow-up. This difference in results may be related to the younger age of the patients with SCI at the time of the interview. In Asian countries, IDC is commonly recommended as a simple and less invasive bladder management technique to treat NLUTD in patients with acute SCI. For patients with chronic SCI, long-term IDC through urethral catheterization or suprapubic cystostomy is commonly accepted as bladder management in real-world practice, especially when the patient has experienced urological complications after CIC. Although IDC can easily induce UTI, urinary calculi, urethral stricture, fistula formation, and other complications, patients with SCI generally do not wish to change their bladder management for bladder emptying when they do not observe any urological complications. Patel et al. reported that patients with SCI may discontinue CIC because of the inconvenience of self-performance, urinary leakage, and further urinary infections. These reasons for discontinuing initial bladder management were similar to those identified in the current study [6]. Patients with chronic SCI may select more suitable bladder management after the acute phase to improve their quality of life and prevent urological complications.

Although patients with SCI with NLUTD cannot completely recover, appropriate and active bladder management can minimize the development of urological complications, and urological procedures or surgeries can improve patients’ satisfaction with their bladder condition. A study indicated that the incidence of UTI was high in new patients with SCI with IDC but could be reduced by removing catheters [13]. Recurrent UTI is the most common urological complication in patients with chronic SCI, as previously reported by Gao et al. [3] and Singh et al. [14]. A lower UTI rate was also noted among patients with SCI who used CIC than among those who had IDC [10,15]. This study has two limitations. First, this study had a cross-sectional study design. Therefore, demonstrating dynamic changes in bladder management was difficult. Second, this study conducted a survey in free clinics settings, which limited the study generalizability to patients who couldn’t attend clinics due to transportation inconvenience. A longitudinal study with a larger sample is warranted.

## 5. Conclusions

Among the 515 patients included in this study, 84.7% were satisfied with their current bladder management, and 55% had changed their initial bladder management at least once after injury. In addition, the participants who had bladder management changes and those who were dissatisfied with their current bladder management had higher urological complication rates than their counterparts. Many patients with SCI with urological complications must change their bladder management to protect their renal function and improve their quality of life; patients who still experience urological complications after changing are generally dissatisfied with their bladder management. However, in the present study, the form of bladder management did not influence patients’ self-reported satisfaction. Proper selection of bladder management in SCI patients needs to be tailored based on patient preference, daily activity level, caregiver support, and the occurrence of UTI. Ultimate success of the SCI bladder management is an art of balancing the patient’s preference and unwanted complication. 

## Figures and Tables

**Figure 1 ijerph-19-17032-f001:**
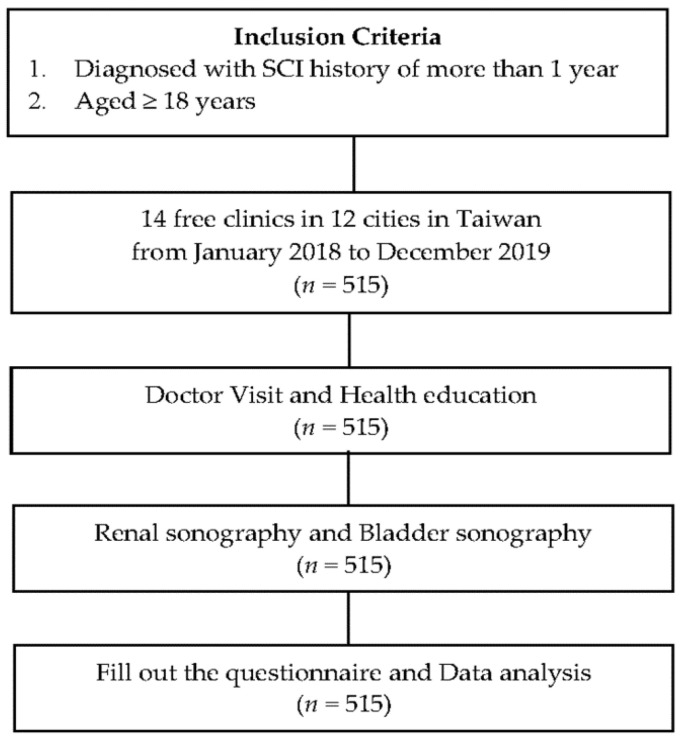
The flow diagram of the study.

**Figure 2 ijerph-19-17032-f002:**
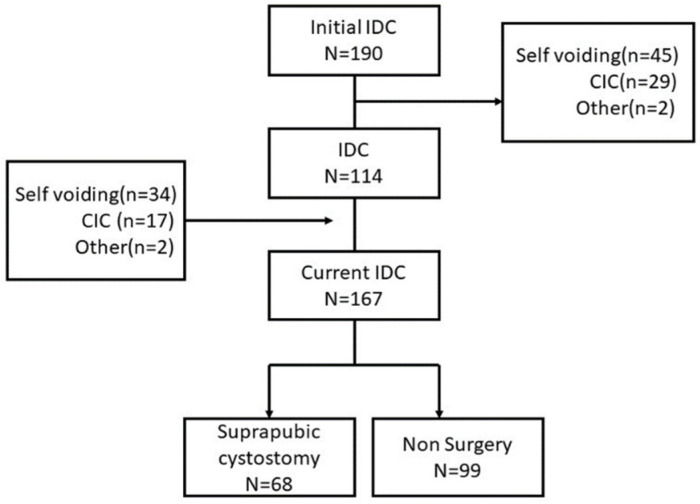
The flow chart of changing bladder management in the initial IDC group.

**Figure 3 ijerph-19-17032-f003:**
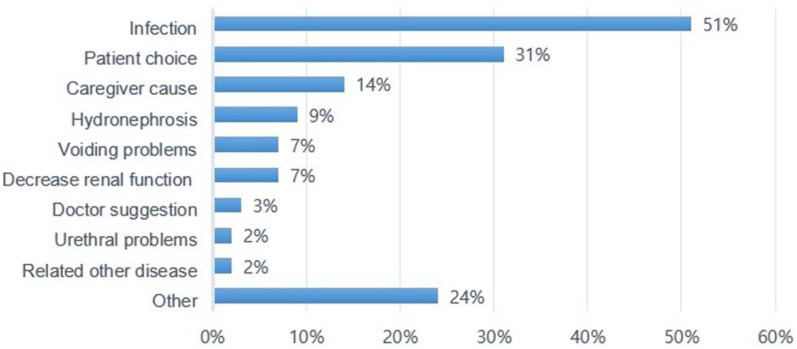
The reasons for changing bladder management (*n* = 283).

**Table 1 ijerph-19-17032-t001:** Demographic and injury characteristics of 515 patients with spinal cord injury.

Characteristics	No. (%)
Sex	
Male	398 (77.3)
Female	117 (22.7)
Age (Mean ± SD)	52.56 ± 13.85
Injury age (year), (Mean ± SD)	32.75 ± 14.53
Time since injury (year), (Mean ± SD)	19.29 ± 12.72
Marital	
Unmarried	208 (40.4)
Married	227 (44.1)
Divorced	60 (11.7)
Widowed	14 (2.7)
Other	6 (1.2)
Injury	
Cervical	235 (45.6)
Thoracic	181 (35.1)
Lumber	95 (18.4)
Sacrum	4 (0.8)
ASIA impairment	
A	248 (48.2)
B	96 (18.6)
C	94 (18.3)
D	68 (13.2)
E	9 (1.7)
Limbs	
Tetraplegia	131 (25.4)
Paraplegia	305 (59.2)
Other	79 (15.3)
Initial bladder management	
Self-voiding	239 (46.4)
IDC	190 (36.9)
CIC	80 (15.5)
Other	6 (1.2)
Current bladder management	
Self-voiding	245 (47.6)
IDC	167 (32.4)
CIC	95 (18.4)
Other	8 (1.6)
Urology Surgery	
No	186 (36.1)
Yes	329 (63.9)
Urology complication	
No	90 (17.5)
Yes	425 (82.5)

Abbreviations: IDC, indwelling catheter; CIC, clean intermittent catheterization; ASIA, American spinal injury association.

**Table 2 ijerph-19-17032-t002:** Patients’ satisfaction with current bladder management (*n* = 515).

Characteristics	Satisfaction No. (%)	Characteristics No. (%)	*p*
Number	436 (84.7)	79 (15.3)	
Sex			
Male	339 (77.8)	59 (74.7)	0.549
Female	97 (22.2)	20 (25.3)	
Age at interview (Mean ± SD)	52.06 ± 14.17	55.37 ± 11.62	0.026 *
Injury age (year), (Mean ± SD)	32.50 ± 14.76	34.08 ± 13.21	0.377
Time since injury (year), (Mean ± SD)	19.55 ± 12.52	21.29 ± 12.91	0.258
Marital			0.254
Unmarried	181 (41.5)	27 (34.2)	
Married	190 (43.6)	37 (46.8)	
Divorced	46 (10.6)	14 (17.7)	
Widowed	13 (3.0)	1 (1.3)	
Other	6 (1.4)	0 (0)	
Injury			0.495
Cervical	201 (46.1)	34 (43.0)	
Thoracic	154 (35.3)	27 (43.2)	
Lumber	77 (17.7)	18 (22.8)	
Sacrum	4 (0.9)	0 (0)	
ASIA impairment			0.607
A	209 (47.9)	39 (49.4)	
B	78 (17.9)	18 (22.8)	
C	81 (18.6)	13 (16.5)	
D	61 (14.0)	7 (8.9)	
E	7 (1.6)	2 (2.5)	
Limbs			0.359
Tetraplegia	116 (26.6)	15 (19.0)	
Paraplegia	254 (58.3)	51 (64.6)	
Other	66 (15.1)	13 (16.5)	
Current bladder management			0.377
Self-voiding	202 (46.3)	43 (54.4)	
IDC	141 (32.3)	26 (32.9)	
CIC	86 (19.7)	9 (11.4)	
Other	7 (1.6)	1 (1.3)	
Urology Surgery			0.905
No	157 (36.0)	29 (36.7)	
Yes	279 (64.0)	50 (63.3)	
Urology complication			0.005 *
No	85 (19.5)	5 (6.3)	
Yes	351 (80.5)	74 (93.7)	

Abbreviations: IDC, indwelling catheter; CIC, clean intermittent catheterization; ASIA, American spinal injury association. * *p* < 0.05 was statistically significant.

**Table 3 ijerph-19-17032-t003:** Patients with or without a change in bladder management of 515 patients with spinal cord injury.

Characteristics	Without Change No. (%)	With Change No. (%)	*p*
Number	232 (45.0)	283 (55.0)	
Sex			
Male	181 (78.0)	217 (76.7)	0.718
Female	51 (22.0)	66 (23.3)	
Age at interview (year), (Mean ± SD)	56.45 ± 12.81	49.37 ± 13.87	0.000 *
Injury age (year), (Mean ± SD)	34.86 ± 16.01	31.01 ± 12.96	0.003 *
Time since injury (year), (Mean ± SD)	21.59 ± 12.99	18.36 ± 12.06	0.004 *
Marital			0.040 *
Unmarried	81 (34.9)	127 (44.9)	
Married	120 (51.7)	107 (37.8)	
Divorced	24 (10.3)	36 (12.7)	
Widowed	6 (2.6)	8 (2.8)	
Other	1 (0.4)	5 (1.8)	
Injury			0.080
Cervical	109 (47.0)	126 (44.5)	
Thoracic	66 (28.4)	115 (40.6)	
Lumber	55 (23.7)	40 (14.1)	
Sacrum	2 (0.9)	2 (0.7)	
ASIA impairment			0.003 *
A	93 (40.1)	155 (54.8)	
B	43 (18.5)	53 (18.7)	
C	53 (22.8)	41 (14.5)	
D	36 (15.5)	32 (11.3)	
E	7 (3.0)	2 (0.7)	
Limbs			0.001 *
Tetraplegia	53 (22.8)	78 (27.6)	
Paraplegia	128 (55.2)	177 (62.5)	
Other	51 (22.0)	28 (9.9)	
Current bladder management			0.000 *
Self-voiding	154 (66.4)	91 (32.2)	
IDC	51 (22.0)	116 (41.0)	
CIC	23 (9.9)	72 (25.4)	
Other	4 (1.7)	4 (1.4)	
Urology Surgery			0.000 *
No	113 (48.9)	73 (25.9)	
Yes	119 (51.3)	210 (74.2)	
Urology complication			0.000 *
No	69 (29.7)	21 (7.4)	
Yes	163 (70.3)	262 (92.6)	
Satisfaction to bladder management			0.195
Very good	50 (21.6)	46 (16.3)	
Good	60 (25.9)	72 (25.4)	
Not bad	95 (40.9)	113 (39.9)	
Dissatisfied	18 (7.8)	38 (13.4)	
Very dissatisfied	9 (3.9)	14 (4.9)	

Abbreviations: IDC, indwelling catheter; CIC, clean intermittent catheterization; ASIA, American spinal injury association. * *p* < 0.05 was statistically significant.

## Data Availability

The data are available from the corresponding author upon reasonable.

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
