# Peer review of "Reasons for Altering Bladder Management and Satisfaction with Current Bladder Management in Chronic Spinal Cord Injury Patients"

_ijerph, 2022, doi:10.3390/ijerph192417032_

Round 1

Reviewer 1 Report

The aim of this study was to assess patient satisfaction with current bladder 56 management and the reasons for altering the bladder management method in patients with chronic SCI.

There are some comments that should be addressed. 

The sentences cannot initiate with numeric. Please revise them.

The major concern is the measurement tool of the study outcomes. It should be clearly defined what scale was used to evaluate the patients' satisfaction, as well as the complications. It should be attached to the manuscript. In the limitation section, the authors mentioned " this study did not include participants who had tetraplegia and had no caregiver or transportation inconvenience that prevented them from attending the free clinical examination", however, in the results section there were cases with tetraplegia. Please clarify this inconsistency. 

Author Response

Review Report –Reviewer 1 (Round 1)

Comments and Suggestions for Authors

The aim of this study was to assess patient satisfaction with current bladder 56 management and the reasons for altering the bladder management method in patients with chronic SCI.

There are some comments that should be addressed.

The sentences cannot initiate with numeric. Please revise them.

Reply: Thank you for your reminder. We have revised the error (line 20).

The major concern is the measurement tool of the study outcomes. It should be clearly defined what scale was used to evaluate the patients' satisfaction, as well as the complications. It should be attached to the manuscript.

Reply: Thank you for your valuable comments. We have added defined what scale was used to evaluate the patients' satisfaction, as well as the complications. The explanations are as below (line 95-98). “Satisfaction was evaluated by one global rating scale on satisfaction, scales are divided into “Not at all Satisfied,” “Partly Satisfied,” “Satisfied,” “More than Satisfied,” “Very Satisfied,” five categories. The urological complication was evaluated that includes 11 items of complication for patients to choose whether they have ever occurred. ”

In the limitation section, the authors mentioned “this study did not include participants who had tetraplegia and had no caregiver or transportation inconvenience that prevented them from attending the free clinical examination", however, in the results section there were cases with tetraplegia. Please clarify this inconsistency.

Reply: Thank you for your reminder. We have rewritten the sentence in limitation and make it clearer. “This study conducted survey in free clinics settings, which limited study generalizability to patients who couldn’t attend clinics due to transportation inconvenience.” (line 263-265)

Reviewer 2 Report

The authors may present details of the change of bladder management under each category. For example, how many patients changed the bladder management from indwelling urethral catheter drainage to intermittent catheterisation and vice versa. Then the authors should present the reason for changing the bladder management. For example, why did a patient stop intermittent catheterisation and resorted to indwelling catheter drainage. Was it due to lack of carers or did the patient develop complication of intermittent catheterisation e.g., urethral bleeding. 

Similarly, the authors should discuss patients with urethral catheter drainage underwent supprapubic cystostomy. If so, when did they undergo suprapubic cystostomy - was it six months after the spinal cord injury or years later. What was the rationale for changing the bladder management?

Thus the research design needs to be changed to capture meaningful medical data. The present report appears to be superficial with no take home message. 

Author Response

Review Report –Reviewer 2 (Round 1)

Comments and Suggestions for Authors

The authors may present details of the change of bladder management under each category. For example, how many patients changed the bladder management from indwelling urethral catheter drainage to intermittent catheterisation and vice versa. Then the authors should present the reason for changing the bladder management. For example, why did a patient stop intermittent catheterisation and resorted to indwelling catheter drainage. Was it due to lack of carers or did the patient develop complication of intermittent catheterisation e.g., urethral bleeding.

Reply: Thank you for your suggestions. We added the description of the reason for changing the bladder management from intermittent catheterization to indwelling catheterization. “Of patients in the initial CIC group, seventeen of them switched to the IDC group, due to recurrent UTI (15, 88%) and lack of helper (2, 12%)( line 168-169). Besides, we have added the figure 2 for details of the initial IDC change of bladder management flow chart and reason. One hundred and ninety people were in the IDC group at the first 6 months, whereas 114 (60%) remained in IDC, 29 (15.3%) changed to CIC and 45 (23.7%) regained control of self-voiding.” (line 169-171). In discussion, we also explains the result (line 221-222 & 229-235)

Similarly, the authors should discuss patients with urethral catheter drainage underwent supprapubic cystostomy. If so, when did they undergo suprapubic cystostomy - was it six months after the spinal cord injury or years later. What was the rationale for changing the bladder management?

Reply: Thank you for your suggestion. We have added description about patients with urethral catheter drainage underwent supprapubic cystostomy in the Result (line 171-174). “Sixty eight (40.7%) of the current IDC group had underwent suprapubic cystostomy surgery(Figure 2). The reasons of the changes were recurrent infection (24, 34.5%), care related factors (26, 38%), and personal factors (18, 26%).”

Thus the research design needs to be changed to capture meaningful medical data. The present report appears to be superficial with no take home message.

Reply: Thank you for your suggestion. In conclusions, we added the result (line 2279-282). We would like to thank you for the extensive assessment of our manuscript, and for important and helpful comments and suggestions. We have taken all the remarks into account, in a manner that is described in detail with our revision. We also thank that, following your suggestions, our manuscript has gained in clarity and hope that the changes made will be considered satisfactory.

Round 2

Reviewer 1 Report

-

Reviewer 2 Report

The authors have tried to improve the manuscript. I accept it. Still, I am not completely satisfied that this manuscript is backed by scientific data. The choice of bladder management should be based upon the urodynamic parameters, findings of ultrasound scan of the urinary tract. We should explain the findings of these tests and the rationale for the choice of bladder management to the patient and the relatives/carers. 

I would plead with the authors that they should go back the drawing board, get more information regarding the medical aspects of these cases and get the input from urologist and spinal cord physician. What was done to address urinary tract infections in persons practicing intermittent catheterisation?  We have been able to continue intermittent catheterisation and reduce the episodes of urine infections by improving personal hygiene, changing the type of catheter, using no touch technique, using prophylactic antibacterial, etc. 

The authors talk about the marital status of the participants. The authors should discuss the clinical significance of each parameter which they report. 

The article requires lot of improvement.